# The Magnitude of Cardiovascular Disease Risk Factors in Seafarers from 1994 to 2021: A Systematic Review and Meta-Analysis

**DOI:** 10.3390/jpm13050861

**Published:** 2023-05-20

**Authors:** Getu Gamo Sagaro, Ulrico Angeloni, Claudia Marotta, Giulio Nittari, Giovanni Rezza, Andrea Silenzi, Gopi Battineni, Francesco Amenta

**Affiliations:** 1Telemedicine and Telepharmacy Center, School of Medicinal and Health Products Sciences, University of Camerino, 62032 Camerino, Italygopi.battineni@unicam.it (G.B.); francesco.amenta@unicam.it (F.A.); 2School of Public Health, College of Health Sciences and Medicine, Wolaita Sodo University, Sodo 138, Ethiopia; 3General Directorate of Health Prevention, Ministry of Health, 00144 Rome, Italy; 4Research Department, International Radio Medical Center (C.I.R.M.), 00144 Rome, Italy

**Keywords:** hypertension, overweight, obesity, smoking, epidemiology, prevalence, seafarers, ships

## Abstract

Objectives: The incidence of acute cardiac events is one of the main reasons for medical consultation, disembarkation, repatriation, and death among seafarers at sea. Managing cardiovascular risk factors, particularly those that can be modified, is the key to preventing cardiovascular disease. Therefore, this review estimates the pooled prevalence of major CVD risk factors among seafarers. Methods: We conducted a comprehensive search of studies published between 1994 and December 2021 in four international databases, namely PubMed/Medline, Scopus, Google Scholar, and Web of Science (WOS). Each study was evaluated for methodological quality using the Joanna Briggs Institute (JBI) critical appraisal tool for prevalence studies. The DerSimonian–Laird random-effects model with logit transformations was used to estimate the pooled prevalence of major CVD risk factors. The results were reported in accordance with the Preferred Items for Systematic Review and Meta-analysis (PRISMA) guidelines. Results: Out of all 1484 studies reviewed, 21 studies with 145,913 study participants met the eligibility criteria and were included in the meta-analysis. In the pooled analysis, the prevalence of smoking was found to be 40.14% (95% CI: 34.29 to 46.29%) with heterogeneity between studies (*I*^2^ = 98%, *p* < 0.01). The prevalence of hypertension, overweight, obesity, diabetes mellitus, and alcohol consumption was 45.32%, 41.67%, 18.60%, 12.70%, and 38.58%, respectively. However, the sensitivity analysis after excluding studies showed a pooled prevalence of hypertension, overweight, obesity, and diabetes mellitus of 44.86%, 41.87%, 15.99%, and 16.84%, respectively. The subgroup analysis demonstrated that smoking prevalence among seafarers had decreased significantly after 2013. Conclusion: This study demonstrated that CVD risk factors, particularly hypertension, overweight, smoking, alcohol consumption, and obesity, are prevalent among seafarers. These findings may serve as a guide for shipping companies and other responsible bodies in order to prevent CVD risk factors among seafarers. PROSPERO Registration: CRD42022300993.

## 1. Introduction

Globally, cardiovascular diseases (CVD) account for the majority of disease burden and are attributed to both modifiable and unmodifiable risk factors [1]. CVD are also the number one cause of death from disease on board among seafarers [2,3]. On board a ship, acute cardiac events are one of the leading causes of medical consultation, disembarkation, repatriation, and mortality among seafarers [4,5,6,7,8]. The risk of cardiovascular events among seafarers is higher than that of the general population [9,10]. This may be due to a variety of reasons, including inadequate treatment, no regular monitoring, no immediate response to the emergency despite its severity, delayed resuscitation action, or work-related stress [11].

It is well known that the working conditions of sailors influence their health. A seafarer’s work is characterized by long working hours, lack of sleep and frequent interruptions in their sleep, as well as staying at sea for extended periods of time, which adversely affects their health [12,13,14]. Due to the particular circumstances of their working environment, seafarers can experience different coping strategies such as unhealthy lifestyles (such as smoking, alcohol consumption, etc.) [13,15]. In addition to physical and psychological stresses, these unhealthy lifestyles contribute to CVD on board ships [16]. In order to prevent CVD, risk factors, particularly those that are modifiable, need to be managed.

Modifiable risk factors, such as tobacco use, heavy alcohol consumption, overweight/obesity, and physical inactivity are highly prevalent among seafarers [9,10,17,18,19]. Additionally, the prevalence of modifiable risk factors varies widely among mariners. In a recent systematic review, the prevalence of modifiable risk factors was reported; for smoking, the prevalence was between 37.3% and 72.3%, for overweight between 27.9% and 66.5%, for high blood pressure between 8.2% and 49.7%, and for diabetes mellitus, it ranged from 3.3% to 9.3% [20]. Another systematic review found that the prevalence of alcohol consumption among seafarers varies widely, from 11.5% to 89.5% [21]. As a result, the data presented on the prevalence of modifiable CVD risk factors among seafarers are inconsistent between studies. Inconsistent data on the prevalence of modifiable risk factors for CVD among seafarers may lead decision-makers as well as researchers to consider different figures based on their preferences and the available information. To date, no studies have been reported on the pooled prevalence of CVD risk factors in seafarers. In order to make evidence-based decisions, an analysis of the pooled prevalence of major risk factors for CVD is essential.

The present study aimed to estimate the prevalence of major CVD risk factors (cigarette smoking, high blood pressure, diabetes mellitus, overweight, obesity, and alcohol consumption) among seafarers by reviewing literature available on the topic and analyzing it with a meta-analysis prevalence approach. The results of this study could help international organizations [e.g., International Maritime Organization (IMO), International Labor Office (ILO), World Health Organization (WHO)], national governments, trade unions, shipping companies, and other decision-makers to develop strategies to improve the control of CVD risk factors on board ships among seafarers.

## 2. Methods

The present systematic review followed the Preferred Items for Systematic Review and Meta-analysis (PRISMA) checklists and diagrams to design and report the results [22], and registered a protocol for this review with the International Prospective Register of Systematic Reviews (PROSPERO) registration number: CRD42022300993).

It is available from https://www.crd.york.ac.uk/prospero/display_record.php?ID=CRD42022300993 (accessed on 8 May 2023).

### 2.1. Research Questions

This study was guided by the following primary research questions: What is the magnitude of major CVD risk factors among seafarers? Does the distribution of CVD risk factors on-board ships differ according to the time period? How does age affect the distribution of CVD risk factors?

### 2.2. Search Strategy and Data Sources

In order to identify relevant studies, we conducted a comprehensive systematic search of the literature according to the Meta-Analysis of Observational Studies in Epidemiology (MOOSE) guidelines [23] and the PRISMA statement [22]. We searched the following databases PubMed/Medline, Scopus, Google Scholar, and Web of Science (WOS) for studies reporting the prevalence of CVD risk factors, specifically smoking, high blood pressure, diabetes mellitus, overweight, obesity, and alcohol use, up to November 2021. Further relevant articles were manually reviewed from the retrieved study reference lists. We applied the following key terms for searching in PubMed, Scopus, and WOS for hypertension: “prevalence”, “proportion”, “magnitude”, “high blood pressure”, “hypertension”, “seafarers”, “onboard ships”, “merchant ships”, and “sailors”. To combine the search terms for each outcome of interest, we used Boolean operators such as “AND” and “OR”. The full search strategy in PubMed and Scopus for the prevalence of hypertension, overweight, obesity, smoking, diabetes mellitus, and alcohol use can be found in Appendix A.

### 2.3. Inclusion and Exclusion Criteria

The following criteria were considered for eligibility: (1) observational studies (cross-sectional, cohort, and case-control); (2) studies reporting on the prevalence of high blood pressure, overweight, obesity, smoking, diabetes mellitus (DM), and alcohol consumption; (3) studies published between 1994 and December 2021; (4) full-text studies written in English. The following studies were not considered in this study: (1) studies that were not peer-reviewed or were unpublished; (2) studies published as abstracts or conference proceedings; (4) qualitative studies; (5) studies with a small sample size (less than 50 study participants); (6) studies published in languages other than English; (7) review studies, i.e., either systematic or narrative reviews.

In this study, six co-authors (G.G.S., U.A., C.M., G.N., A.S., and G.B.) carried out a literature search and selected the studies independently based on the inclusion criteria. While conducting the literature search and selecting the studies, the two senior co-authors (G.R. and F.A.) resolved any disagreements between the authors.

### 2.4. Data Extraction and Outcome Variables

After selecting studies, the variables extracted from each study were the first author’s name, publication year, number of cases or reported prevalence, sample size, and study design. These data were entered into an Excel spreadsheet. The primary outcome of the present study was the pooled prevalence of CVD risk factors (high blood pressure, smoking, diabetes, overweight, obesity, alcohol consumption). The five authors (G.G.S., U.A., C.M., G.N., and G.B.) extracted data and compared the results. Any discrepancies between the results were resolved through discussion.

### 2.5. Quality Assessment

The Joanna Briggs Institute (JBI) critical appraisal tool was used to assess the methodological quality of the studies [24]. The critical appraisal tool contains ten items that were used to evaluate the methodological quality of studies reporting prevalence data (see Appendix A). A critical appraisal was performed prior to data extraction. Despite the fact that there are four possible responses to each question in the critical appraisal tool (“yes”, “no”, “unclear” or “not applicable”), there is no indication in the document as to how the assessment tool should be interpreted quantitatively in order to rank the studies as low or high quality. Some studies, however, used the mean scores to measure the quality of studies [25,26].

In the present study, the quality of the studies was evaluated using agreed-upon category scores for each study. As a result, the studies were categorized into low, medium, and high quality based on scores ranging from 0 to 10. The studies scoring between 0 and 4 were considered low-quality, the studies scoring between 5 and 6 were considered medium-quality, while studies scoring seven and above (7–10) were considered high-quality.

### 2.6. Statistical Analysis

The data were entered into a Microsoft Excel spreadsheet version 2019 and analyzed using R-software (Version 4.1.1, The R Foundation for Statistical Computing, Vienna, Austria) [27]. We used the metaprop()functions from R package meta [28] for prevalence and summary meta-analysis and we employed also the *escalc()*, *rma()*, and *predict()* functions from R package metafor [29] along with different arguments to calculate individual effect size (i.e., proportions) and their corresponding sampling variance estimation. A DerSimonian–Laird random-effects model with logit transformations was used to estimate the pooled prevalence of CVD risk factors (high blood pressure, smoking, overweight, obesity, diabetes mellitus, and alcohol consumption) [30]. A random-effect model was used to adjust observed variability [31]. The pooled proportion of each CVD risk factor, considered in the present study with a 95% CI, was generated and visualized using a forest plot.

Begg’s and Egger’s tests were performed to detect the potential publication bias [32,33]. Heterogeneity between studies was assessed using the Cochran’s Q test [34] and I2 test statistics [35]. The degree of heterogeneity was considered as low, moderate, and high based on *I*^2^ values of less than 25%, 25% to 75%, and more than 75%, respectively [36]. A univariate meta-regression analysis was conducted based on publication years and sample size to estimate their impact on the prevalence of each CVD risk factor. We also performed a sensitivity analysis using the “Leave-one-out” analysis with a built-in function. Once the outliers were identified, we re-estimated the summary effect (i.e., pooled prevalence) by omitting outliers. Subgroup analyses were also performed according to the year of publication.

## 3. Results

### 3.1. Study Characteristics

In total, 1484 records were identified using our search strategy, of which 954 records were excluded because of duplicates. The title and abstract screening excluded 495 articles. The remaining 35 full-text articles were evaluated. Among the 35 full-text papers reviewed, 21 studies with 145,913 study participants met the eligibility criteria and were included in the meta-analysis [9,10,17,37,38,39,40,41,42,43,44,45,46,47,48,49,50,51,52,53,54]. Figure 1 shows the entire process of finding, selecting, and including studies (Figure 1).

The studies included in the present systematic review and meta-analysis were conducted between 1994 and 2021. All the included studies were cross-sectional studies (out of 21 studies, 3 were retrospective analyses of cross-sectional studies). The characteristics of the selected studies are summarized in Table 1 along with their methodological quality assessment (Table 1).

### 3.2. Operational Definition of Outcome Variables in the Included Studies

Four studies defined high blood pressure/hypertension according to the current World Health Organization (WHO) criteria [55]: systolic blood pressure (SBP) ≥ 140 mmHg and/or diastolic blood pressure (DBP) ≥ 90 mmHg and/or taking antihypertensive drugs. In the remaining four studies, high blood pressure was defined as a SBP ≥ 130 mmHg and/or a DBP ≥ 85 mmHg or antihypertensive medication. Smoking status (n = 16) was assessed using a self-reported questionnaire. It was subsequently verified by asking questions such as about the duration of use, age at onset, and the number of cigarettes per day. Overweight (n = 12) and obesity (n = 12) were defined according to current WHO criteria using body mass index (BMI) [56]: 25.0 kg/m^2^
< BMI < 30.0 kg/m^2^, and ≥30 kg/m^2^, respectively, and the BMI was also calculated as weight in kilograms (kg) divided by height in meters (m) squared [Weight (kg)/Height (m)^2^]. Regarding diabetes mellitus, it was defined as follows: fasting plasma glucose level > 110 mg/dL (n = 1), fasting blood glucose level > 126 mg/dL or blood glucose level 2 h after eating > 200 mg/dL (n = 1), fasting plasma glucose ≥ 5.6 mmol/L or previously diagnosed type two diabetes (n = 1), and fasting glucose level ≥ 110 mg/dL and/or anti-diabetic medication use (n = 2).

### 3.3. Prevalence of CVD Risk Factors

#### 3.3.1. Prevalence of Smoking

Sixteen studies were selected with a total of 11,511 study participants [9,10,17,38,39,40,41,42,43,45,46,47,48,49,50,52]. In general, the prevalence of smoking varied greatly between the 16 studies, ranging from 23.85% [46] to 67.18% [10]. The pooled prevalence of smoking among seafarers was found to be 40.14% (95% CI: 34.29% to 46.29%), with a high and statistically significant heterogeneity (*I*^2^ = 98%, *p* < 0.01). We took into account the year of publication as a subgroup analysis of smoking prevalence. Thus, seven studies were published between 1994 and 2012, and nine studies were published between 2013 and 2020. The publication year was then categorized into two groups: 2013 and after (2013–2021), and before 2013 (1994–2012). As a result, the pooled proportion of smoking was 34.43% (95% CI: 25.90% to 44.11%, *I*^2^ = 98%, *p* < 0.01) during the 2013 year of publication and after, and 47.85% (95% CI: 41.24% to 54.52%, *I*^2^ = 95%, *p* < 0.01) before 2013. There was a significant decline in smoking prevalence (*p* < 0.01) in 2013 and subsequent years compared to before 2013 (34.43% vs. 47.85%) (Figure 2).

The findings of univariate meta-regression analysis showed that sample size had no impact on the prevalence of smoking among seafarers [QM (test of moderators) (1) = 0.956, *p* = 0.328]. The year of publication had an impact on the observed prevalence of smoking in seafarers. In fact, there was an association between the prevalence of smoking and year of publication (QM(1) = 9.648, *p* < 0.001) as well as the significant slope coefficient [−0.059, *Z*(14) = −3.106, *p* = 0.002]. The *R*^2^ for the publication year shows that 20.47% of the true heterogeneity in the presented effect size can be explained by the year of publication.

The sensitivity analysis indicated no evidence of outliers among the included studies for smoking prevalence (see Appendix A).

#### 3.3.2. Prevalence of High Blood Pressure

The pooled prevalence of high blood pressure among seafarers was 45.32% (95% CI: 36.98% to 53.93%) with significant heterogeneity between the studies (*I*^2^ = 96%, *p* < 0.01) (Figure 3). Overall, eight studies were identified with a total of 3554 study participants [9,17,38,41,42,43,46,48]. The prevalence of high blood pressure varied between the selected studies, ranging from 21.23% [41] to 70.42% [43]. As for the prevalence based on the year of publication, the overall proportion of prevalence of high blood pressure (HBP) was 51.74% (95% CI: 37.90% to 65.32%) after the 2013 year of publication, and 39.02% (95%CI: 29.85 to 49.03%) before 2013 (Figure 3).

A sensitivity analysis was performed and two outlier studies were identified [41,43], which influenced the pooled estimate of high blood pressure (see Appendix A). After omitting the outlier studies, the overall prevalence of high blood pressure was 44.86% (95%CI: 43.03% to 46.71%) (Appendix A), which indicates that the pooled prevalence decreased slightly after removing the outlier studies.

#### 3.3.3. Prevalence of Overweight

Overall, twelve studies reporting the overweight prevalence with a total of 136,710 participants were selected for the meta-analysis [10,17,37,38,41,42,44,46,48,51,53,54]. In selected studies, the prevalence of overweight varied from 28.09% [42] to 51.51% [10]. The pooled prevalence of overweight among seafarers was 41.67% (95% CI: 39.16% to 44.22%, *I*^2^ = 85%, *p* < 0.01). Five of the twelve studies analyzed for combined prevalence were published before 2013 and the remaining seven studies were published after 2013. As a result, the pooled proportion of overweight before the 2013 year of publication was found to be 40.71% (95% CI: 33.67% to 48.16%), and it was 42.15% (95% CI: 39.46% to 44.88%) after 2013. Thus, the prevalence of overweight increased slightly after 2013 compared to before 2013, although the difference was not statistically significant (*p* = 0.72) (Figure 4).

The meta-regression analysis indicated that both sample size [QM(1) = 0.209, *p* = 0.647] and publication year [QM(1) = 1.495, *p* = 0.222] were not significantly associated with the proportion of high blood pressure. The sensitivity analysis identified two studies that had influenced the overall prevalence of overweight (Appendix A). After removing the two outlier studies [10,42], the pooled prevalence of overweight was 41.87% (39.88% to 43.89%) with heterogeneity between studies (*I^2^ =* 70%, *p* < 0.01) (see Appendix A).

In two studies, the prevalence of overweight was assessed by age group [37,44]. As a result, the overall prevalence of overweight among seafarers aged 16–24 years was 25.64% (95% CI: 18.43% to 34.48%), and 48.84% (95% CI: 43.66% to 54.04%) among those aged 45–66 years (Table 2). The results of our study demonstrated that overweight in seafarers increases significantly with age (x^2^(2) = 18.46, *p* < 0.001).

#### 3.3.4. Prevalence of Obesity

We included 12 studies reporting data on obesity in the present meta-analysis, with a total of 136,710 subjects [10,17,37,38,41,42,44,46,48,51,53,54]. In selected studies, obesity prevalence varied widely, ranging from 8.55% [48] to 61.08% [42]. The pooled prevalence for obesity was 18.60% (95% CI: 13.24% to 25.48%, *I*^2^ = 99%, *p* < 0.01). As for the years of publication, seven studies were published after 2013, and five studies before 2013. Regarding publication-year-specific prevalence, the combined proportion of obesity after 2013 was 15.14% (95% CI: 11.93% to 19.03%), and 23.84% (95%CI: 11.61% to 42.72%) before 2013. The magnitude of obesity before 2013 was higher than after 2013 (23.84% vs. 15.14%), but the difference was not statistically significant (*p* = 0.23) (Figure 5).

Among the 12 studies included in the meta-analysis for the prevalence of obesity in seafarers, only 2 studies [37,44] reported the obesity prevalence stratified by the age group of the seafarers. As a result, the pooled prevalence of obesity among seafarers aged from 16 to 24 years was 5.10% (95%CI: 2.05% to 12.10%), and 26.74% (95%CI: 20.13% to 34.59%) in seafarers aged between 45 and 66 years (Table 2). According to these findings, obese seafarers aged 45 to 66 years had a higher prevalence, and the difference between the age groups was statistically significant as well (X^2^ (2) = 16.37, *p* < 0.001).

According to the univariate meta-regression analysis results, both sample size [QM(1) = 0.344, *p* = 0.557] and publication year [QM(1) = 0.280, *p* = 0.596] were not significantly associated with the proportion of obesity. By conducting sensitivity analysis, one outlier study was identified (Appendix A), which influenced the pooled prevalence of obesity. After omitting the outlier study [42], the overall prevalence of obesity was found to be 15.99% (95% CI: 12.88% to 19.68%), with substantial heterogeneity between studies (*I*^2^ = 95%, *p* < 0.01) (Appendix A).

#### 3.3.5. Prevalence of Diabetes Mellitus

A total of 1519 participants were included in five studies that investigated the prevalence of diabetes mellitus [38,41,42,43,48]. The overall proportion of diabetes mellitus in the five studies included in the meta-analysis varied from 3.30% [41] to 23.08%. The pooled prevalence for diabetes mellitus was 12.70% (95%CI: 7.88% to 19.85%, *I*^2^ = 92%, *p* < 0.001). Among the five studies, three studies were published before 2013 and the remaining two studies were published after 2013. The combined proportion for diabetes mellitus from papers published after 2013 was 20.10% (95%CI: 15.60% to 25.51%, *I*^2^ = 63%, *p* < 0.01). However, the prevalence of DM before 2013 was 7.62% (95%CI: 1.84% to 26.69%, *I*^2^ = 95%, *p* < 0.01) (Figure 6).

Using sensitivity analysis, one outlier study was identified (Appendix A). After excluding the outlier study [41], the combined prevalence of diabetes mellitus (DM) was 16.84% (11.75% to 23.53%, *I*^2^ = 86%, *p* < 0.01) (Appendix A). Consequently, the overall prevalence of DM was increased after omitting the outlier study.

#### 3.3.6. Prevalence of Alcohol Consumption

A total of ten studies with 8093 participants provided data on alcohol consumption prevalence [9,38,40,43,45,46,47,49,50,52]. Overall, among the ten studies included in the meta-analysis, alcohol consumption proportion varied widely, ranging from 8.05% [40] to 82.56% [9]. A pooled prevalence of alcohol consumption was 38.56% (95%CI: 19.68% to 61.69%, *I*^2^ = 100%, *p* < 0.001) (Appendix A). In terms of the publication years, seven studies were published after 2013, and three studies were published before 2013. Taking into account publication-year-specific prevalence, alcohol use prevalence after 2013 was 33.42% (95% CI: 17.11% to 54.98%), and 51.32% (95%CI: 6.48% to 94.14%) before 2013. The prevalence of alcohol use before 2013 was higher than after 2013 (51.32% vs. 33.42%), but the difference was not statistically significant (*p* = 0.61) (Appendix A).

We conducted a sensitivity analysis in order to identify outliers among the included studies in the meta-analysis. The sensitivity analysis, however, did not reveal any evidence of outliers among the included studies (see Appendix A).

### 3.4. Publication Bias

As for the publication bias, neither Egger’s (*p* = 0.690) nor Begg’s (*p* = 0.571) tests were statistically significant, indicating that no publication bias occurred.

## 4. Discussion

In the present systematic review and meta-analysis, we estimated the magnitude of CVD risk factors (smoking, high blood pressure, overweight, obesity, diabetes mellitus, and alcohol consumption) among seafarers. We synthesized the findings of 21 published studies with a total of 145,913 study participants between 1994 and 2021 that met the eligibility criteria to estimate the prevalence of major CVD risk factors. We considered the literature from 1994 and onwards in our study search, since we did not find any relevant studies on CVD risk factor prevalence before 1994 based on our preliminary search of different worldwide databases when looking for studies on seafarers. In addition, we searched for peer-reviewed studies on CVD risk factor prevalence from 1994 until 31 December 2021, in the databases we selected because this study began in January 2022. As for the methodological quality assessment of the included studies, 14.3% (n = 3), 38.1% (n = 8), and 47.6% (n = 10) of the studies were of low, medium, and high methodological quality (Table 1). Among the major CVD risk factors considered in this study, high blood pressure (HBP) was the most common risk factor (45.32%), with high and significant heterogeneity (*I*^2^ = 96%, *p* < 0.01). After the sensitivity analysis, HBP (44.86%) was also the main common CVD risk factor compared to the other risk factors included in this study. In a study conducted among seafarers, high blood pressure was identified as a leading cause of cardiovascular disease and accounted for 89% of all CVD diagnosed between 2010 and 2018 on board ships [57].

For the purpose of comparing the magnitude of CVD risk factors, we created two groups based on the study period (before and after 2013). The year 2013 was utilized as a cut-off point because different initiatives related to seafarers’ health were implemented or amended in 2013 and thereafter [58,59]. Therefore, we were interested in studying changes in the magnitude of common CVD risk factors over time. According to the sub-group analysis, the pooled prevalence of HBP was higher after the 2013 year of publication than before 2013 (51.74% vs. 39.02%), indicating an increase in the magnitude of HBP on board ships. After 2013, different measures were undertaken to improve the health of seafarers at sea. As an example, the 2010 International Convention on Standards of Training, Certification, and Watchkeeping for Seafarers (STCW) [58] and the 2006 Maritime Labor Convention (MLC) entered into force on 20 August 2013 [60]. The MLC 2006 outlined numerous health services for seafarers, including physical examination, health monitoring, mandatory limits on board ships, and lifestyle management. Nevertheless, CVD and its risk factors, most notably HBP, were estimated to be more prevalent among seafarers after 2013. Perhaps this is due to the ineffective implementation of measures specified by STCW 2010 and MLC 2006 in relation to the health protection of seafarers after 2013. On the other hand, the IMO, shipping companies, and other responsible bodies need to pay close attention to the implementation of the above conventions and health services for seafarers.

A study conducted on board ships reported that overweight and obesity increased, by 6.70 and 16.75 times, respectively, the risk of high blood pressure among seafarers [61]. Other studies also reported that the prevalence of high blood pressure increases with augmented body mass index, job duration at sea, working hours per week, and age of seafarers [9,62]. The application of specific interventions targeting risk factors such as weight management, limiting daily and weekly working hours in accordance with the MLC 2006 convention, and the regular monitoring and application of prevention measures targeting older seafarers would help to reduce the risk of high blood pressure on board ships.

In this study, we found overweight to be the second most prevalent modifiable risk factor for CVD in seafarers (41.67% with *I*^2^ = 85%, *p* < 0.01). We performed the sensitivity analysis and omitted two outlier studies among the studies included in the meta-analysis for overweight prevalence. We then re-estimated the prevalence of overweight (41.87%) and it was slightly higher than the estimated prevalence before sensitivity analysis. Based on the subgroup analysis, the prevalence of overweight was found to be higher after the 2013 year of publication compared to before 2013 (42.15% vs. 40.71%). We also stratified the proportion of overweight by age group, and accordingly the prevalence of overweight significantly increased with an increase in the age of seafarers. The results obtained are consistent with previous studies conducted among seafarers [61,62]. The possibilities of physical activity on board ships are limited due to the working conditions and the lack of access to a gymnasium on some merchant ships at sea [37]. Consequently, overweight becomes one of the most prevalent risk factors for CVD and can cause relevant health problems at sea. To reduce body weight and the likelihood of CVD, preventive measures such as nutrition management, physical training, and gyms on board ships should be considered. The popularity and diffusion of gyms are increasing on modern cargo ships. It is imperative to follow a physical activity plan under the supervision of a physician and/or trainer in order to maximize the benefits of physical activity to prevent CVDs. During the pre-employment examination, body weight and BMI should be considered as relevant recruitment criteria for seafarers. 

Smoking was found to be the third most common modifiable risk factor for CVD among seafarers in the present study (40.14%). Our study demonstrated that smoking was significantly reduced after 2013 compared to years before 2013 (34.43% vs. 47.85%). This could be due to the application of certain mandatory limits related to smoking on board ships and the awareness of the consequences of smoking among seafarers after 2013. Similarly, Pougnet R and his colleagues [20] reported that smoking prevalence was significantly lower in the 2000s compared to the 1990s (45.4% vs. 61.3%, *p* < 0.01). We encourage applying effective preventive measures and mandatory limits for other common risk factors also, such as high blood pressure, overweight, and alcohol consumption, in order to reduce their prevalence. In general, smoking prevalence is still higher among seafarers. Mandatory limits such as prohibiting smoking in some ship areas should be enforced to reduce the proportion of this phenomenon. Health promotion interventions such as conducting smoking cessation campaigns and raising awareness of the consequences of smoking would improve the control of cigarette smoking on board ships. A study conducted on board ships indicated that level of education is significantly correlated with smoking [52]. Hence, specific campaigns directed at the people more vulnerable in this respect should be considered. Another modifiable CVD risk factor prevalent among seafarers was alcohol consumption [38.56% (95%CI: 19.68% to 61.69%)]. Pooled alcohol consumption was lower after 2013 than before 2013, although the difference was not statistically significant (33.42% vs. 51.32%, *p* = 0.61). This reduction in alcohol use prevalence may be attributed to the update of preventive measures for alcohol and drug abuse by the International Maritime Organization (IMO) in 2010. For example, the International Maritime Organization (IMO) updated the International Convention on Standards of Training, Certification, and Watchkeeping for Seafarers (STCW Convention) in 2010 in order to address the issue of alcohol and drug abuse among seafarers [63]. The magnitude of alcohol consumption on board is still high, and responsible bodies, including the International Maritime Organization (IMO), shipping companies, and other stakeholders, need to develop mitigation strategies to reduce the prevalence of alcohol consumption among seafarers as it is a critical safety issue that should be addressed. The IMO should also evaluate whether the amended STCW convention regarding alcohol use has been fully implemented.

Obesity and diabetes mellitus were also important risk factors for CVD among seafarers. The estimated prevalence of obesity and diabetes mellitus was 18.60% and 12.70%, respectively. However, the sensitivity analysis, after omitting outliers, showed that the combined prevalence of obesity and diabetes mellitus was 15.99% and 16.84%, respectively. We found that the prevalence of obesity increased with the increasing age of seafarers. In addition, the highest prevalence of obesity was observed among older sailors [26.74%, with significant heterogeneity between studies (*I*^2^ = 94%, *p* <0.001)]. Some shipping companies have taken body weight, particularly obesity, into account in their recruitment criteria. In the pre-employment examination of Danish seafarers, a BMI of 40 kg per square meter or more results in exclusion from working on board ships [64]. Norway too have introduced some limitations for the recruitment of seafarers with a BMI of 35 kg/m^2^ or above [37]. Obesity not only increases the risk of diabetes mellitus, high blood pressure, and the burden of CVD, but also renders seafarers unfit for work on board ships. Seafarers often experience a sedentary lifestyle on board. Consequently, it is important to encourage regular exercise, to plan physical activity and health education through telemedicine, and to provide smart offline mobile applications to guide seafarers in improving their physical activity. Lifestyle changes such as physical activity, a healthy diet, and the availability of a gymnasium on board ships could positively influence the body weight of seafarers. The prevalence of diabetes mellitus has increased in parallel with the increase in work experience at sea, age, and weekly working hours. In other words, long job duration at sea, long working hours per week, and older age increase the risk of high blood glucose levels in seafarers [62].

### 4.1. Strengths and Limitations

This is the first review to estimate a pooled prevalence in the context of major risk factors for cardiovascular disease among seafarers at sea. We registered this review protocol initially with the International Prospective Register of Systematic Reviews (PROSPERO) and adhered to the PRISMA guidelines when designing, conducting, and reporting our findings to ensure the validity of the methods used.

Even though most of the included studies were of a low risk of bias, this review found substantial heterogeneity among the included studies, which affected the quality of the overall evidence. Perhaps this is due to poor methodological approaches employed by the various studies. There are a few studies on the health of seafarers centered on their cardiovascular diseases, and data on CVD risk factor prevalence are in general limited. Almost all of the studies included in this review were cross-sectional and some of them had poor methodological quality. In addition, we did not find studies that stratified the prevalence of modifiable CVD risk factors by the rank, nationality, and workplace of seafarers; therefore, we did not take into consideration rank, nationality, and worksite differences in the distribution of prevalence of risk factors for CVD. The magnitude of hypercholesterolemia was not considered in this study due to a lack of studies, despite being one of the major CVD risk factors. We, therefore, encourage future studies to take into account these variables and evaluate their prevalence in a pooled analysis. Despite the above limitations, the estimated proportion of the most common risk factors of CVD is relevant for evidence-based decision making, and for the development of prevention initiatives and control strategies to mitigate the burden of CVD at sea.

### 4.2. Implications for Practice

Modifiable risk factors are precursors to cardiovascular disease, which results in morbidity, mortality, and the need to divert ships from their intended course at sea. Seafarers experience more cardiovascular events than the general population. Moreover, the prognosis after CVD at sea is also worse than ashore [5]. Cardiovascular diseases have received less attention among maritime seafarers in comparison to the general population, although the magnitude of cardiovascular diseases at sea is growing. However, medical emergencies on all types of ships were caused most often by cardiovascular diseases [65]. It is estimated that the shipping industry incurs approximately EUR 253 million in costs as a result of ships diverting from their courses due to medical emergencies, and the total cost for the whole shipping industry is estimated to be around EUR 760 million [66]. The average cost of a ship diverting due to medical emergencies is EUR 2200 per hour [66].

In order to improve seafarers’ health and reduce the economic and other consequences due to cardiac emergencies on board ships, modifiable risk factors should be managed. Our study is the first review reporting the pooled prevalence of modifiable risk factors, and highlighting the high prevalence of modifiable CVD risk factors among seafarers. It is also pointed out in this review that overweight and obesity are prevalent among seafarers, and this poses a safety hazard on board a ship. An overweight or obese seafarer may find it difficult to perform emergency operations such as using the emergency exits or climbing onto a rescue boat. Therefore, this review informs telemedical maritime assistance services (TMAS) physicians who provide teleconsultation services to seafarers by providing prevention advice or scheduled counseling on lifestyle changes in order to reduce modifiable risk factors, especially high body mass index. As a result, seafarers with high BMIs (25 kg/sqm and over) should be advised through telemedicine to engage in lifestyle measures, including exercise and dietary modification.

Furthermore, the results of our study alert telemedicine case managers or specially trained maritime officers who work with seafarers on board ships to monitor their blood pressure and blood glucose levels regularly. The working conditions of seafarers make monitoring regular blood pressure, blood glucose levels, and other lipid profile tests on board ships very challenging. However, thanks to telemedicine technologies, it is now possible to track seafarers’ physiological parameters regularly and report the data to the TMAS doctors. Consequently, the TMAS doctors will contact the telemedicine case manager or the person responsible for healthcare services on board, or, if possible, they will contact the user directly. A real-time consultation via telemedicine is recommended for patients with elevated blood pressure or abnormal parameters.

Moreover, our study provides information to shipping companies to implement policies prohibiting smoking because smoking is not only a health problem but also a risky habit and a cause of fires on ships. The review findings, in general, urge shipping companies, and other responsible bodies, such as the IMO, MLC, and maritime health policymakers to focus on prevention programs in order to reduce modifiable CVD risk factors on board ships. We recommend that future studies take into account the causes of the modifiable risk factors on board ships.

## 5. Conclusions

The present study has demonstrated that seafarers have a high prevalence of CVD risk factors, particularly high blood pressure (45.32%), overweight (41.67%), smoking (40.14%), obesity (18.60%), and alcohol consumption (38.58%). This review found substantial heterogeneity between the included studies, although most of the included studies had a low risk of bias, which affected the certainty of the overall evidence. The present study also indicated that the pooled prevalence of overweight and obesity increased along with seafarers’ age. The findings of this review will help the IMO, shipping companies, and other stakeholders to develop and implement telemedicine prevention strategies that address the common CVD risk factors considered in this study.

## Figures and Tables

**Figure 1 jpm-13-00861-f001:**
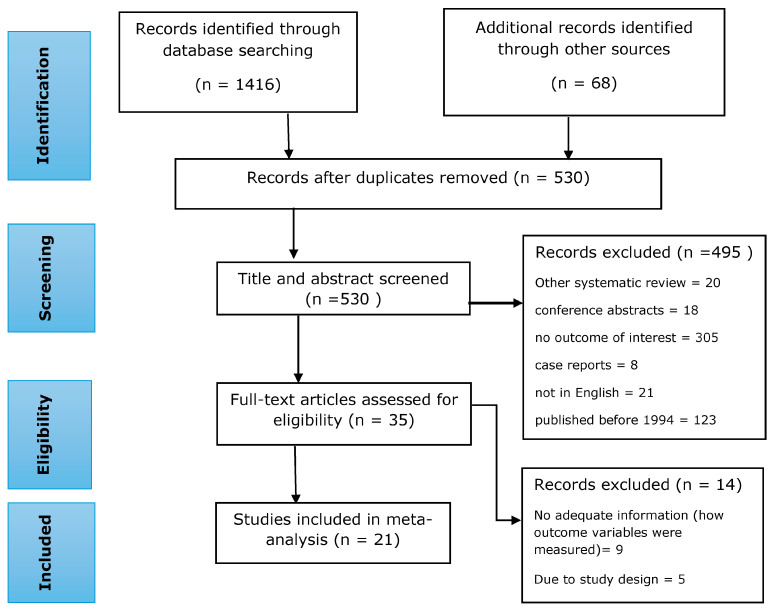
A flow diagram showing the process of study searching, selection, and inclusion in the present systematic review and meta-analysis.

**Figure 2 jpm-13-00861-f002:**
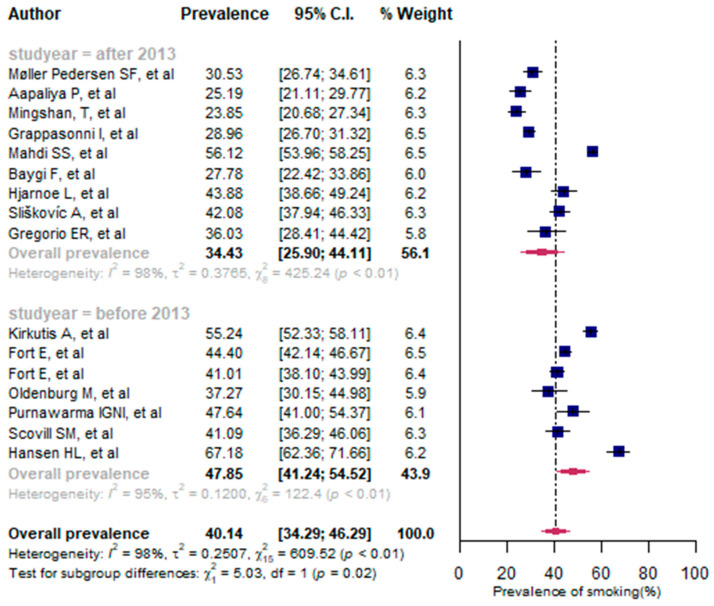
A forest plot of the prevalence (%) of smoking among seafarers using a random-effects model.

**Figure 3 jpm-13-00861-f003:**
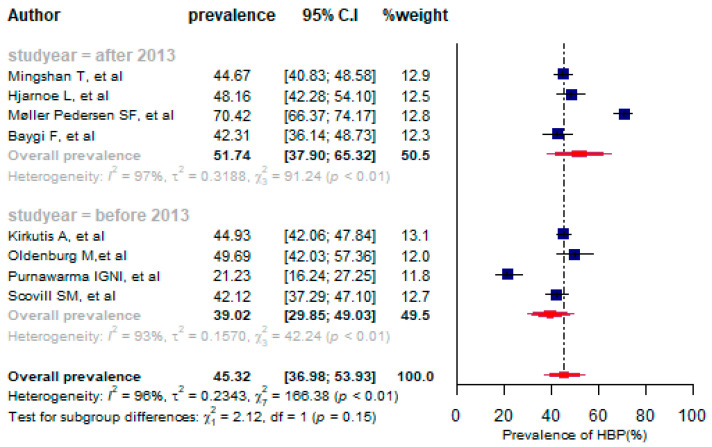
A forest plot of the prevalence (%) of high blood pressure among seafarers using a random-effects model.

**Figure 4 jpm-13-00861-f004:**
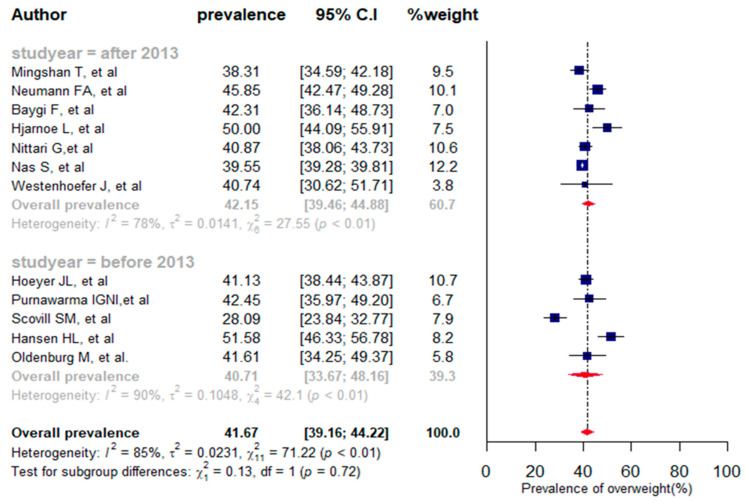
A forest plot of the prevalence (%) of overweight among seafarers using a random-effects model.

**Figure 5 jpm-13-00861-f005:**
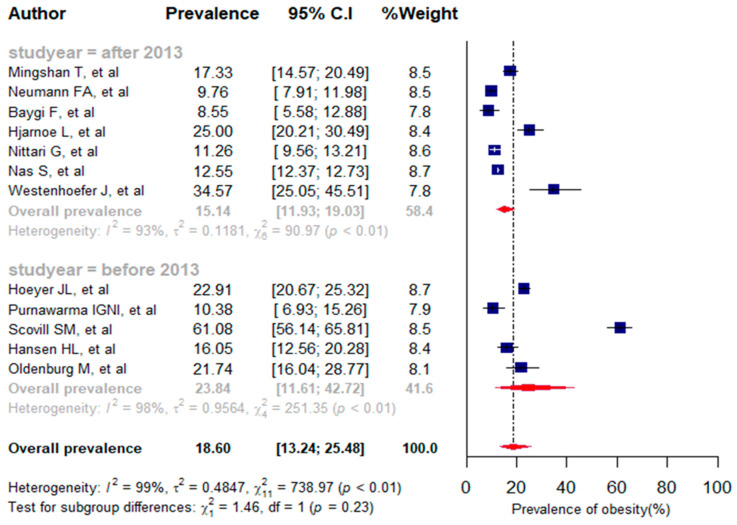
A forest plot of the prevalence (%) of obesity among seafarers using a random-effects model.

**Figure 6 jpm-13-00861-f006:**
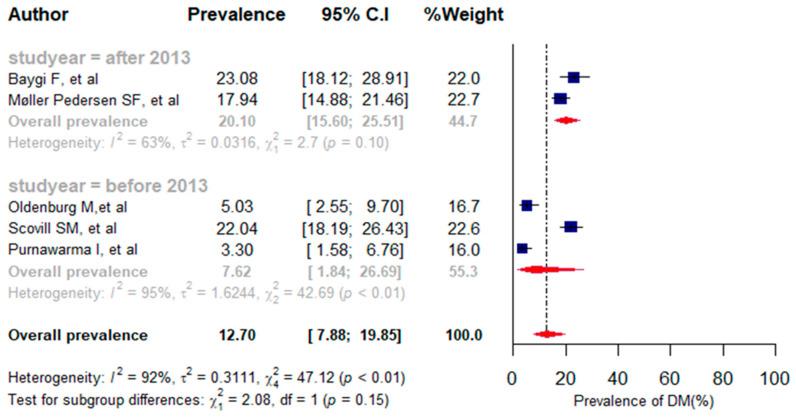
A forest plot of the prevalence (%) of diabetes mellitus among seafarers using a random-effects model.

**Table 1 jpm-13-00861-t001:** Characteristics of selected studies for systematic review and meta-analysis.

Author Name and Year	Study Design	Sample Size	Prevalence (%)						Quality Score
			HBP	Smoking	Diabetes Mellitus	Overweight	Obesity	Alcohol Use	
Hansen, H.L., et al., 1994 [10]	Cross-sectional	390	NA	67.2	NA	51.6	16.1	NA	7
Kirkutis, A., et al., 2004 [9]	Cross-sectional	1135	44.9	55.2	NA	NA	NA	82.6	9
Hoeyer, J.L., et al., 2005 [37]	Retrosp. Cross-sectional	1257	NA	NA	NA	41	22.9	NA	8
Oldenburg, M., et al., 2008 [38]	Cross-sectional	161	49.7	37.3	5	41.6	21.7	73.9	9
Fort, E., et al., 2009 [39]	Cross-sectional	1847	NA	44	NA	NA	NA	NA	6
Fort, E., et al., 2010 [40]	Cross-sectional	1068	NA	41.4	NA	NA	NA	8.0	9
Purnawarma, I., et al., 2011 [41]	Cross-sectional	212	21.2	47.6	3.3	42.5	10.4	NA	7
Scovill, S.M., et al., 2012 [42]	Cross-sectional	387	42	41	22	28	61	NA	7
Møller Pedersen, S.F., et al., 2013 [43]	Cross-sectional	524	70.4	30.6	17.9	NA	NA	18.6	4
Hjarnoe, L., et al., 2014 [17]	Cross-sectional	272	48	44	NA	50	25	NA	5
Nas, S., et al., 2014 [44]	Retrosp. Cross-sectional	131,152	NA	NA	NA	39.6	12.5	NA	4
Aapaliya, P., et al., 2015 [45]	Cross-sectional	385	NA	25.2	NA	NA	NA	14.3	4
Mingshan, T., et al., 2016 [46]	Cross-sectional	629	44.7	23.9	NA	38.3	17.3	71.9	7
Mahdi, S.S., et al., 2016 [47]	Cross-sectional	2060	NA	56.11	NA	NA	NA	11.5	5
Baygi, F., et al., 2016 [48]	Cross-sectional	234	42.3	27.8	23.1	42.5	8.6	NA	6
Gregorio, E.R., et al., 2016 [49]	Cross-sectional	136	NA	36.0	NA	NA	NA	79.4	5
Sliškovíc, A., et al., 2017 [50]	Cross-sectional	530	NA	42.0	NA	NA	NA	41.7	6
Westenhoefer, J., et al., 2018 [51]	Cross-sectional	81	NA	NA	NA	40.7	34.6	NA	5
Grappasonni, I., et al., 2019 [52]	Cross-sectional	1478	NA	28.9	NA	NA	NA	19.5	9
Nittari, G., et al., 2019 [53]	Retrosp.cross-sectional	1155	NA	NA	NA	40.8	11.2	NA	5
Neumann, F.A., et al., 2021 [54]	Cross-sectional	820	NA	NA	NA	45.8	9.8	NA	8

NA = Not Assessed/not assessed according to WHO/IDF criteria [High Blood pressure (HBP), overweight, obesity, Diabetes Mellitus (DM)], and alcohol consumption.

**Table 2 jpm-13-00861-t002:** Prevalence of overweight and obesity in terms of age among seafarers.

	Age Group (Years)	Pooled Prevalence (95% CI)	*I*^2^ (*p*-Value)
Overweight	16–24	25.64% (18.43–34.48)	77% (0.04)
	25–44	41.49% (37.25–45.86)	80% (0.03)
	45–66	48.84% (43.66–54.04)	84% (0.001)
Obesity	16–24	5.10% (2.05–12.10)	87% (0.001)
	25–44	15.14% (10.30–21.69)	95% (0.001)
	45–66	26.74% (20.13–34.59)	94% (0.001)

## Data Availability

All relevant data are included in the article or provided as Appendix A. All extracted data are available upon request from the corresponding author.

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
