# Peer review of "The Magnitude of Cardiovascular Disease Risk Factors in Seafarers from 1994 to 2021: A Systematic Review and Meta-Analysis"

_jpm, 2023, doi:10.3390/jpm13050861_

Round 1

Reviewer 1 Report

Overall, this is a concise manuscript. The current study described the modifiable CVD risk factors among seafarers. The novelty of the study is average, and the results provide a certain perspective on the guidance of risk management. 

Sufficient information about the previous study findings is presented for readers to follow the present study rationale and procedures.

The methods are slightly confusing and needed further clarification. The Method section did not sufficiently describe in enough detail how the literature review and analysis were conducted.

The discussion section is relevant but needed further organization and evidence support. 

The other significant differences in lifestyles of seafarers - food structure and sleep deficiency are not included in the analysis. Also, the ethnicity of the population could be a covariate and contribute to the CVD risks. These cofounders might limit the generality of the results, therefore, should be further discussed and/or mentioned/included in the limitation section.

Minor editing of English language required.

Author Response

May 9, 2023

Responses to the reviewer’s #1 comments

We thank the reviewer for taking the time to carefully read our manuscript and for providing us with detailed comments and suggestions that have helped us to improve the manuscript we are resubmitting.

Please find below our response to each of the comments and suggestions:

With best regards!

Getu Gamo Sagaro (Ph.D.)

Reviewer #1

  1. “The methods are slightly confusing and needed further clarification. The Method section did not sufficiently describe in enough detail how the literature review and analysis were conducted.”

We are grateful to the reviewer for his/her valuable comments and suggestions. To address these comments, we have clearly outlined the steps we followed in reviewing the literature from the literature search to the analysis of the extracted data. Moreover, we described how we managed outlier studies in the section on data analysis. The revised version of the manuscript includes information regarding the prevalence of alcohol consumption. We would appreciate it if the reviewer could review the revised manuscript.

  1. “The discussion section is relevant but needed further organization and evidence support.”

The discussion section has been revised in response to the reviewer's comments.

  1. “The other significant differences in lifestyles of seafarers - food structure and sleep deficiency are not included in the analysis. Also, the ethnicity of the population could be a covariate and contribute to the CVD risks. These cofounders might limit the generality of the results, therefore, should be further discussed and/or mentioned/included in the limitation section.”

We would like to thank the reviewer for his/her valuable comments and suggestions. In the revised version of the manuscript, these points are addressed in the limitation of the study section.

Reviewer 2 Report

I would like to thank the authors for their manuscript which reports about the prevalence of modifiable CVDs risk factors among seafarers.I found the manuscript very good and timely done. Nonetheless, there are a number of shortcomings:

Major comments:

The authors mentioned the aim of the study is the evaluation of modifiable CVDs risk factors however hypertension and diabetes mellitus aren’t modifiable risk factors. They are classified as non-modifiable risk factors or co-morbidities that can increase the risk of developing CVD ) according to the classification of the international guidelines.

On the other hand, the prevalence of major CVDs modifiable risk factors such as hypercholesterolemia and alcohol consumption were not included in the analysis.

I suggest that the authors include hypercholesterolemia and alcohol consumption in their analysis and change the aim of the study to evaluating 'different' CVDs risk factors.

The title should also be changed as well ..as it is currently (Prevalence of Modifiable Risk Factors.

The manuscript need to be revised accordingly.

Minor comments: 

- Could you please clarify why data was collected in this time period (1994 and December 2021)

- Could you please clarify why you specifically used 2013 as the cut-off in the comparison of your data?

- Were there any differences between the prevalence of diabetes mellitus type 1 and type 2 in the assessed data? please add this information to your results section.

Please add references to the following sentences:

Line 47-48: The risk of cardiovascular events among seafarers is higher than that of the general population

Line 48-51: This may be due to a variety of reasons, including inadequate treatment, no regular monitoring, no immediate response to the emergency despite its severity, delayed resuscitation action, or work-related

Line 55-59: Due to the particular circumstances of their working environment, seafarers would experience different coping strategies such as unhealthy lifestyles 56 (such as smoking, alcohol consumption, etc.). In addition to physical and psychological 57 stresses, these unhealthy lifestyles contribute to CVD onboard ships. In order to prevent  CVD, risk factors, particularly those that are modifiable, need to be managed.

Author Response

May 9, 2023

Responses to the reviewer’s#2 comments

We thank the reviewer for taking the time to carefully read our manuscript and for providing us with comments and suggestions that have helped us to improve the manuscript we are resubmitting.

Please find below our response to each of the comments and suggestions:

With best regards!

Getu Gamo Sagaro (Ph.D.)

Reviewer #2

  1. “The authors mentioned the aim of the study is the evaluation of modifiableCVDs risk factors however hypertension and diabetes mellitus aren’t modifiable risk factors. They are classified as non-modifiable risk factors or co-morbidities that can increase the risk of developing CVD ) according to the classification of the international guidelines.”

First of all, we would like to thank the reviewer for his/her insightful comments and suggestions. We have accepted the comments and revised them in our revised version of the manuscript.

  1. On the other hand, the prevalence of major CVDs modifiablerisk factors such as hypercholesterolemia and alcohol consumption were not included in the analysis. I suggest that the authors include hypercholesterolemia and alcohol consumption in their analysis and change the aim of the study to evaluating 'different' CVDs risk factors. The title should also be changed as well ..as it is currently (Prevalence of Modifiable Risk Factors. The manuscript need to be revised accordingly.”

In response to these comments, we have included in our revised version of the manuscript the alcohol consumption prevalence on pages 16 and 17, and other points raised by the reviewer have been revised accordingly. Hypercholesterolemia, however, was not considered due to a lack of studies, but it was mentioned clearly in the limitations of the study. Consequently, we would appreciate if the reviewer could visit our revised version of the manuscript.

  1. “Could you please clarify why data was collected in this time period (1994 and December 2021)”

We appreciate the reviewer's comments, and we have explained why we considered these time periods on page 17, beginning at line number 5.

  1. “Could you please clarify why you specifically used 2013 as the cut-off in the comparison of your data?”

The reasons for using 2013 as the cut-off point are explained on page 17, paragraph 2.

  1. “Were there any differences between the prevalence of diabetes mellitus type 1 and type 2 in the assessed data? please add this information to your results section.”

Our study did not consider diabetes prevalence separately by type 1 and type 2, and it was also not our interest. We appreciate the reviewer's understanding.

  1. “Line 47-48:The risk of cardiovascular events among seafarers is higher than that of the general population”

In the revised version of the manuscript, the reference has been added.

  1. “Line 48-51:This may be due to a variety of reasons, including inadequate treatment, no regular monitoring, no immediate response to the emergency despite its severity, delayed resuscitation action, or work-related.”

The reference has been added.

  1. “Line 55-59:Due to the particular circumstances of their working environment, seafarers would experience different coping strategies such as unhealthy lifestyles 56 (such as smoking, alcohol consumption, etc.). In addition to physical and psychological 57 stresses, these unhealthy lifestyles contribute to CVD onboard ships. In order to prevent  CVD, risk factors, particularly those that are modifiable, need to be managed.”

The source is cited in the revised version of the manuscript.

Round 2

Reviewer 2 Report

Thank you very much for your effort

You successfully addressed all the recommended changes 

Congratulations on this amazing work